# Perfusion Air Culture of Precision-Cut Tumor Slices: An Ex Vivo System to Evaluate Individual Drug Response under Controlled Culture Conditions

**DOI:** 10.3390/cells12050807

**Published:** 2023-03-04

**Authors:** Meng Dong, Kathrin Böpple, Julia Thiel, Bernd Winkler, Chunguang Liang, Julia Schueler, Emma J. Davies, Simon T. Barry, Tauno Metsalu, Thomas E. Mürdter, Georg Sauer, German Ott, Matthias Schwab, Walter E. Aulitzky

**Affiliations:** 1Dr. Margarete Fischer-Bosch Institute of Clinical Pharmacology and University of Tübingen, 70376 Stuttgart, Germany; 2Department of Gynecology and Obstetrics, Robert Bosch Hospital, 70376 Stuttgart, Germany; 3Department of Bioinformatics, Biocenter, University of Würzburg, 97074 Würzburg, Germany; 4Charles River Germany GmbH, Am Flughafen 12-14, 79108 Freiburg, Germany; 5Bioscience, Early Oncology, AstraZeneca, Cambridge CB2 0AA, UK; 6Institute of Computer Science, University of Tartu, 51009 Tartu, Estonia; 7Department of Clinical Pathology, Robert Bosch Hospital, 70376 Stuttgart, Germany; 8Departments of Clinical Pharmacology, Pharmacy and Biochemistry, University of Tübingen, 72076 Tübingen, Germany; 9Cluster of Excellence iFIT (EXC2180) “Image-Guided and Functionally Instructed Tumor Therapies”, University of Tübingen, 72076 Tübingen, Germany; 10Department of Oncology, Robert Bosch Hospital, 70376 Stuttgart, Germany

**Keywords:** precision-cut tumor slices, perfusion culture, tumor microenvironment, ovarian tumor, individual drug responses, mouse xenografts, preclinical model, personalized medicine

## Abstract

Precision-cut tumor slices (PCTS) maintain tissue heterogeneity concerning different cell types and preserve the tumor microenvironment (TME). Typically, PCTS are cultured statically on a filter support at an air–liquid interface, which gives rise to intra-slice gradients during culture. To overcome this problem, we developed a perfusion air culture (PAC) system that can provide a continuous and controlled oxygen medium, and drug supply. This makes it an adaptable ex vivo system for evaluating drug responses in a tissue-specific microenvironment. PCTS from mouse xenografts (MCF-7, H1437) and primary human ovarian tumors (primary OV) cultured in the PAC system maintained the morphology, proliferation, and TME for more than 7 days, and no intra-slice gradients were observed. Cultured PCTS were analyzed for DNA damage, apoptosis, and transcriptional biomarkers for the cellular stress response. For the primary OV slices, cisplatin treatment induced a diverse increase in the cleavage of caspase-3 and PD-L1 expression, indicating a heterogeneous response to drug treatment between patients. Immune cells were preserved throughout the culturing period, indicating that immune therapy can be analyzed. The novel PAC system is suitable for assessing individual drug responses and can thus be used as a preclinical model to predict in vivo therapy responses.

## 1. Introduction

Solid tumors are often considered as abnormal organs not only composed of the tumor cells but also the surrounding tumor microenvironment which mainly contains fibroblasts, immune cells, blood vessels, lymphatic vessels, and the extracellular matrix [1,2]. The tumor microenvironment plays a very important role in tumor development and resistance to drug treatment. The functional and physical interaction of the tumor microenvironment with cancer cells plus the variations of the vascular networks within the tumor contribute to inter- and intratumoral heterogeneity and ultimately influence clinical outcomes [3,4]. Therefore, for personalized medicine in precision oncology, it is crucial that a preclinical model captures the complexity and heterogeneity of tumor biology ex vivo to individually predict in vivo therapy of tumors. Precision-cut tumor slices (PCTS) of 200–300 µm thickness maintain both the three-dimensional architecture and tumor heterogeneity, in addition to preserving the native microenvironment concerning different cell types and the extracellular matrix [2]. There have been a growing number of publications using tumor slices as a model to study the tumor microenvironment and address the response of drug treatment [5,6,7,8,9,10,11].

Davies et al. [12] described a standardized workflow for systematic comparison of different tumor slice cultivation methods and showed that the cultivation of tumor slices requires organotypic support materials and atmospheric oxygen to maintain the viability and structure of the tumor slices. The Millipore filter (MF) support culture system under atmosphere was found as one of the promising systems, but it is still associated with significant temporal and loco-regional changes in protein expression such as estrogen receptor (ER) and hypoxia-inducible factor α (HIF1α) in slices of the MCF-7 cell line-derived xenograft (CDX). The development of loco-regional heterogeneity during slice culture makes the data interpretation more complicated in the studies especially with pharmacological perturbation [12,13]. Due to the lack of functional vasculature in the tumor slices, the diffusion of oxygen, nutrients, and drugs are influenced by several variables. Still, it is not yet established in common tumor slice cultures to combine possible perfusion systems to mimic the vasculature on both sides of the tumor slice and further optimize oxygen and nutrients supply and drug distribution [12,14].

Recently, we have developed a perfusion air culture (PAC) system which integrates a perfusion system to partially mimic the vasculature in tumor slices with a continuous and controlled oxygen, medium, and drug supply. In the PAC system, precision-cut tumor slices (PCTS) were kept in between two organotypic supports fixed in a special chamber and placed inside of a 50 mL tube with air exchange capacity housed in a standard CO_2_-incubator. Cotton meshes were used as organotypic supports to cover both sides of the tumor slice in the system. Due to their material structure, the cotton fibers can function as capillaries to supply medium containing nutrients and drugs to the tumor slice. Meanwhile, the relatively big open pores of the cotton mesh allow the oxygen to be easily diffused into the tumor slice from both sides during cultivation. It has been stated that the key parameters governing tissue viability are organotypic filter supports and oxygen levels [12]. The PAC system fulfills these criteria and can supply the oxygen, nutrients, and drugs from the same direction to the tumor slice imitating the in vivo situation. For the in vivo tumor, viable tumor cells were not observed at distances greater than 160 μm from blood vessels [15]. In addition, it was demonstrated that the oxygen can only be diffused to a distance of about 100 µm from a blood vessel [16,17]. In the PAC system, the two cotton meshes supported on both sides of the tumor slice work to mimic the vasculature in a tumor and guarantee the equal distribution and diffusion of oxygen, nutrients, and drugs from both sides of the tumor slices into the deeper cell layers.

Tumor tissue is complex and dynamic, and there are interactions between cancer cells and immune cells in the tumor [1]. In recent years, there has been an increasing interest in using tumor slice cultures as a model to study the tumor immune microenvironment and the immune response of the tumor to the drug treatment [10,18]. Immune cell survival after slice culture in pancreatic cancer has been reported [10]. The immune cells in the slices responded predictably to an immuno-modulator and anti-programmed death- ligand 1 (PD-L1) checkpoint inhibitor blockade [18]. To test whether the PAC system is also suitable to assess functional immune response in tumor slices, we analyzed the immune cells of tumor slices before and after cultivation. The patient-specific immune cells and their composition were preserved throughout the culture period in the PAC system. Increased expression of PD-L1 was observed in the tumor slices after cisplatin treatment.

In this study, we introduce a novel PAC system to culture the tumor slices. It overcomes the problems of the static MF culture system, better represents the complexities of tumor biology, and facilitates the homogenous and controlled supply of oxygen, nutrients, and drugs. It allows the long-term culture of tumor tissue and the analysis of therapy response, including immune therapy, and is thus suitable for individual testing of drug efficacy to predict patient response and enhance drug selection for clinical trials.

## 2. Materials and Methods

Perfusion air culture system construction: The designed chambers (Figure 1) were printed by a 3D printer (Printrbot Simple Metal, Printbot, Lincoln, CA, USA) with polylactic acid (PLA) using the stock thermoplastic extruder. The software used for printing the STL file was Cura Ultimaker. The printed chambers were immersed in 70% ethanol for 15 min, followed by drying out under the cell culture hood for sterilization. The chamber consists of two main components (a and b, Appendix A), which can be assembled very easily under sterile conditions using a click system. Both components together form the holder for the organotypic supports (component c in Appendix A) for the tumor slices. The filter papers (Whatman™ Cellulose Blotting Papers, Grade GB003, #10426892 Cytiva, Marlborough, MA, USA), which are shown as component d in Appendix A, with the property of diffusion were fixed in the chambers (components a and b) and served as a reservoir for the nutrient fluid. The tumor slices were kept in between two organotypic supports and fixed in the chamber (Appendix A). The organotypic supports used in this study were cotton mesh (ES-Kompressen, #2050040 HARTMANN, Heidenheim, Germany, for xenograft tumor slice cultivation and aluderm^®^ Kompressen #KR03029, SÖHNGEN, Taunusstein, Germany, for human ovarian tumor slice cultivation). The decellularized porcine intestine scaffold which was kindly provided by Prof. Heike Walles (Otto-von-Guericke-University, Magdeburg, Germany) was used for the long-term cultivation of human ovarian tumor slices in the PAC system. The chamber was settled vertically inside of a 50 mL tube with air exchange capacity (TubeSpin^®^ Bioreactor 50 with Septum, TPP, Trasadingen, Switzerland). A needle went through the lid of the tube and was inserted in the top of the chamber. The needle was connected to a silicone tube with an inside diameter (ID) of 0.5 mm and an outside diameter (OD) of 2.5 mm. The 1 m-long gas-permeable silicone tube was further connected to a syringe pump for delivery of the medium (Figure 1). The commercially available syringe pump allows precise, low-speed perfusion that can be matched to blood perfusion rates in capillaries (e.g., 50–100 µL/hour; Figure 1). Cultivation was performed in a regular cell culture incubator at 37 °C and 5% CO_2_ under atmospheric oxygen (21% oxygen) conditions.

Cell-line-derived xenograft (CDX) tumor samples: The local committees approved all of the animal facilities and handling protocols on the ethics of animal experiments, as required in each country, and adhered to the European Convention for Protection of Vertebrate Animals used for Experimental Purposes (Directive 2010/63/EU). Experiments with mice performed at AstraZeneca were compliant with the UK Animals (Scientific Procedures) Act, which is consistent with EU Directive 2010/63/EU and had undergone internal ethical review. At Charles River Germany GmbH, experiments carried out with mice were scrutinized by the Committee on the Ethics of Animal Experiments of the regional council (Regierungspräsidium Freiburg, Abt. Landwirtschaft, Ländlicher Raum, Veterinär- und Lebensmittelwesen).

The breast CDX MCF-7 was derived by subcutaneous injection of 5 × 10^6^ MCF-7 cells (ATCC-HTB-22) per 0.1 mL in 50:50 basal media Matrigel (#356234, BD Biosciences, San Jose, CA, USA) into the left flank of male SCID mice (SCID/CB17). The animals were implanted with 0.5 mg/21 day 17β oestradiol pellets (Innovative Research of America) one day before cell implant. The lung CDX NCI-H1437 was derived by subcutaneous injection of 5 × 10^6^ NCI-H1437 cells (ATCC-CRL-5872) into the flank of 4–6-week-old NMRI nu/nu mice. All of the cell lines were routinely (every 3 months) checked for mycoplasma contamination and the master stock of the cells was authenticated using STR analysis before injected being into the mice.

Tumors from mouse xenografts were harvested when the volume reached between 0.4–1 cm^3^. The animals were euthanized by cervical dislocation and tumors were excised. A small sample of the tumor was either snap-frozen using liquid nitrogen or fixed in 10% neutral buffered formalin immediately after resection. Tumors were placed into ice-cold MACS Tissue Storage Solution (#130-100-008, Miltenyi Biotec, Bergisch Gladbach, Germany) before tissue slicing.

Primary human ovarian tumors: Sterile fresh tissue was obtained during debulking surgery in case of ovarian cancer first detected by a frozen section and confirmed by final histological examination at the Robert Bosch Hospital, Stuttgart. Immediately after surgical resection, the tumor tissue was maintained in ice-cold MACS Tissue Storage Solution (#130-100-008, Miltenyi Biotec) until use. The procedure had been approved by the local ethics committee (397/2016BO1) and informed consent from all participating subjects was obtained.

Tumor slice preparation and cultivation: Preparation of tumor slices was performed as described previously [12]. The tumors were mounted onto the magnetic specimen holder of a Leica VT1200S vibrating blade microtome using cyanoacrylate adhesive. The tumor slices were prepared at a thickness of 250 μm (MCF-7 CDX and H1437 CDX) and 280 μm (primary OV) using the precision-cut vibratome. The slices were visually inspected whilst being cut to ensure the tissue was not compressed or torn, resulting in an inconsistent slice thickness. Generally, 15–25 slices can be obtained from one tumor depending on the size and condition of the tumor tissue. MCF-7 CDX slices were cultivated in DMEM (#31053-028, Gibco, Grand Island, NY, USA); H1437 CDX and primary OV slices were cultivated in RPMI 1640 (# F1215, Biochrom AG, Berlin, Germany). The medium was supplemented with glutamine (2 mM; Gibco), penicillin (100 U/mL) and streptomycin (100 μg/mL; Gibco), and 10% fetal bovine serum (FBS; #10082, Gibco).

Cultivation was performed at 37 °C and 5% CO_2_ in a humidified atmosphere under atmospheric oxygen (21% oxygen) conditions. The tumor slices were maintained on a Millipore filter (Millicell Cell Culture inserts, #PICM ORG 50, pore size 0.4 μm, Merck Millipore, PTFE, Merck KGaA, Darmstadt, Germany) with an air-liquid interface in a six-well plate with 1.5 mL medium under the filter in each well and one drop of medium on the top of each slice. The tumor slices were alternatively cultured on the self-made perfusion air culture (PAC) system illustrated in Figure 1. The slices were harvested at different time points. The snap-frozen samples were either collected for RNA isolation (three to four slices per condition) or fixed in 10% neutral buffered formalin for immunohistochemistry (IHC) (one slice per condition). Overall, at least 12–15 slices were analyzed per tumor. The tissue fixed immediately after surgical resection was defined as the in vivo sample. The tissue after the slicing process and before tumor slice cultivation was defined as the d0 sample. The fixed slices were embedded in paraffin in vertical orientation as published in Davies et al., 2015 [12].

Drug treatment: Tumor slices from both xenografts and primary human ovarian tumors were treated with cisplatin (Teva^®^, Ulm, Germany) in both MF and PAC systems. With a final concentration of 13 µM, cisplatin was applied in 1.5 mL medium in the MF system with a drop of medium containing cisplatin on the top of the tumor slices. In the PAC system, the cisplatin in the medium was prepared in a 10 mL syringe and continually applied to the tumor slices through a silicone tube. In order to minimize the influence of tumor heterogeneity, the control and treated slices were always adjacent slices in the tumor.

Immunohistochemical staining: The fixed tumor slices were cut into 4 μm serial sections by Rotary Microtome (Leica RM2255, Wetzlar, Germany). The paraffin sections were stained with hematoxylin (#1.09253.0500, Merck KGaA, Darmstadt, Germany) and eosin (R03040, Merck KGaA) (H&E) for histopathological examination. IHC was carried out by standard protocols as previously described. Briefly, the sections were deparaffinized in Microclear and rehydrated in graded ethanol followed by use of a Dako Envision Kit (#K5007, Agilent Technologies, Glostrup, Denmark) according to the manufacturer’s manual. Epitope retrieval was achieved in a steam heater for 30 min with either citric acid buffer pH 6 or Tris/EDTA buffer pH9 (Agilent Technologies, Dako). The primary antibodies were as follows: Ki67 (Clone MIB-1, Dako), hypoxia-inducible actor 1 α (HIF1α, #610959, BD Biosciences), cleaved-caspase 3 (CC3, #9661, Cell Signaling Technology, Danvers, MA, USA), an estrogen receptor (ER, Clone 6F11, #PA1051, LeicaBond), phospho-histone H2A.X (γH2AX, #2577, Cell Signaling Technology), CD8 (SP16, Cell Marque, Rocklin, CA, USA), CD4 (SP35, Cell Marque), CD68 (Kp-1, Cell Marque), FOXP3 (236A/E7, eBioscience), PD-1 (NAT105, Cell Marque), and PD-L1 (E1L3N, Cell Signaling Technology). The antibodies were visualized using 3,3′-diaminobenzidine (DAB) chromogen and counterstained with hematoxylin. Images were taken by an Olympus slide scanner VS120.

Multiplex immunohistochemical staining: The tissue sections were prepared, deparaffinized, rehydrated, subjected to heat-induced epitope retrieval, and incubated with primary and secondary antibodies as described for immunohistochemical staining. The antibodies were visualized using fluorescent tyramide. The process of epitope retrieval and staining was repeated sequentially for different primary antibody and fluorescent tyramide combinations. The following tyramide dyes were used: CF^®^488 (#92171, Biotium, Fremont, CA, USA), CF^®^555 (#96021, Biotium), and CF^®^640R (#92175, Biotium). After all of the staining steps, the sections were mounted using a DAPI-containing mounting medium (EverBrite™, Biotium). Images were acquired using a Leica TCS SP8 confocal microscope.

Quantification of immunohistochemical staining: The quantification of immunohistochemical staining images was performed in whole tissue sections using the computer-aided image analysis software Tissue Studio from Definiens and QuPath. These programs allow quantification of the positively stained cell numbers in user-defined regions of interest (ROIs). Within these ROIs, algorithms were used that detect nuclei, membranous, and cytoplasmic staining. The mean percentages of positively stained tumor cells from IHC quantification were calculated across tumor types from at least three independent experiments. The Wilcoxon matched-pairs test was used for statistical analysis in GraphPad Prism (GraphPad Software, San Diego, CA, USA). Loco-regional changes in biomarker expression across different areas of the tumor slices were quantified by splitting the tumor slice longitudinally into three layers using ROIs (MF: Filter side, Middle, Air side; PAC: Air side-1, Middle, Air side-2). The percentage difference (|Difference|/Average × 100%) in the positively stained cells between the two outer layers was calculated (MF: Filter side and Air side; PAC: Air side-1 and Air side-2). The average was defined as the average of positively stained cells of the three layers in the tumor slice. Tumor slices with a percentage difference of >20% were defined as having a gradient for this biomarker. The detailed workflow for determining the biomarker expression gradient is shown in Appendix A.

High-throughput TaqMan-based qPCR Fluidigm: Snap-frozen tumor slices were used for total RNA extraction. Lysing matrix D tubes were used to prepare tissue lysates in the FastPrep sample preparation system (MP Biomedicals, Santa Ana, CA, USA). Subsequently, RNA extraction was carried out using an RNeasy Mini Kit (Qiagen, Hilden, Germany) and on-column DNase (Qiagen) digestion was performed to eliminate genomic DNA contamination. M-MLV reverse transcriptase (Promega, Madison, WI, USA) was used to generate the cDNA. Expression analysis was performed on the BioMark HD System (Fluidigm, South San Francisco, CA, USA) according to the manufacturer´s instructions. The TaqMan assays were purchased from Applied Biosystems as previously described [12].

Gene expression analysis: Gene expression qPCR data were normalized to a housekeeping gene and converted to log2 values. Differentially expressed genes were found using the limma package in R statistics environment [19]. Tumor number was included in the linear model to take heterogeneity between the tumors into account. Genes with an FDR-corrected p-value less than 0.05 and a log-fold change of at least 1 were considered as significant.

Euclidean distance (square root of the sum of square differences) was used to measure the difference of the slice models from the in vivo situation. The function removeBatchEffect from limma R package was used to remove the effect of the tumor number before calculating Euclidean distances and completing principal component analysis (PCA) [20]. PCA was completed using the prcomp function in base R. Genes containing any missing values were removed before calculating principal components. The scatterplot showed the first two components (those with the largest and second-largest variance).

## 3. Results

### 3.1. The Perfusion Air Culture (PAC) System Facilitates the Cultivation and Drug Treatment of Tumor Slices under Controlled Conditions

In the perfusion air culture (PAC) system, the tumor slices were kept in between two organotypic supports and fixed in a special chamber allowing continuous perfusion with the medium and drugs. The printed PLA chambers for the PAC system were all used once for each experiment. The PLA material and the 15 min in 70% ethanol sterilization process for the chambers did not show toxicity to the tumor cells (data not shown). The chamber was settled vertically inside a 50 mL tube with air exchange capacity and connected to a syringe pump via a silicone tube (Figure 1). The system was placed in a cell culture incubator at 37 °C and 5% CO_2_ under atmospheric oxygen conditions. The medium pumped out from the syringe went through a 1 m-long highly gas-permeable silicone tube with a 0.5 mm inner and 2.5 mm outer diameter to allow optimal diffusion of oxygen to the culture medium. The flow rate can be adjusted according to the culture conditions or experimental purpose. A low-speed flow rate of 2 mL per day (83.3 µL/h) was used in this study. With such a low perfusion rate, the medium in the highly gas-permeable silicone tube can be well saturated with 21% oxygen conditions at 37 °C in the cell culture incubator before it reached the tumor slices. Therefore, the oxygen concentration of the medium was maintained at the same level of 21% oxygen independent of the samples and experiments. The medium flows through the needle and passes the organotypic supports on both sides of the tumor slices, which offers an air-liquid interface on both sides of the tumor slices. This allows constant exposure to drugs via the medium. The organotypic supports can also be replaced with other materials such as scaffolds from a porcine intestine according to the tissue type and experimental purpose. As a comparison, the tumor slices were also cultured with the Millipore filter (MF) system. In this system, the tumor slices were placed on the filter membrane in the six-well plate with 1.5 mL medium under the filter in each well (Figure 1b). Two or three tumor slices can be cultured together on one MF. One drop of medium was added on the top of each slice to keep it moist without drying it out. In the PAC system, different organotypic supports can be used. After testing several different materials, we used a cotton mesh and scaffold from a porcine intestine for the tumor slice culture (Figure 1d) in this study. The cotton mesh is a highly absorbent gauze. The gauze was first cut into strips to fit the PAC system before use. The scaffold from the porcine intestine is a biological vascularized scaffold which can be used as a dynamic 3D matrix system. The structure of the scaffold is shown in Figure 1d as H&E staining.

### 3.2. Slices Cultured with a Perfusion Air Culture (PAC) System Maintain the Morphology and Viability of the Tumor Tissue for More than 7 Days

Mouse xenografts (MCF-7, H1437) and primary human ovarian tumor (primary OV) tissues were cultured in the PAC system and compared to the commonly used static Millipore filter (MF) culture system. After 3 days of cultivation in the PAC system, tumor slices of MCF-7 xenografts, a breast cancer model, showed similar morphology and biomarker expression (ER, HIF1α, Ki67, and γH2AX) to the original tissues (Figure 2b). The ER expression of the MCF-7 tumor slices was significantly higher in the PAC system compared to the MF system, and more close to the in vivo situation (Figure 2d). The MCF7 xenografts tumor slices were cultured for up to 7 days with only minor changes in viability and morphology (Figure 3a,c). The tumor slices of H1437 xenografts, a lung cancer model, were stained for Ki67, HIF1α, γH2AX, and cleaved-caspase 3 (CC3) to investigate cell proliferation, oxygen supply, DNA damage and apoptosis. The H1437 slices showed similar morphology and biomarker expression after 3 days of culture compared to the day 0 (d0) non-cultivated tumor slices (Figure 4). Primary OV tissues cultured with decellularized scaffolds from porcine intestines together with cotton meshes as organotypic supports maintained their morphology and biomarker expression for up to 8 days (Figure 5a).

### 3.3. Loco-Regional Changes in Biomarkers Induced in the Filter Support Culture System Were Overcome with the Perfusion Air Culture (PAC) System

In order to evaluate loco-regional changes across slices during culture, formalin-fixed tumor slices were vertically embedded in paraffin blocks as previously published [12]. The tumor slices were objectively divided into three layers longitudinally according to the shape of the tumor slices (MF: Air side, Middle, Filter side; PAC: Air side-1, Middle, Air side-2) using the QuPath software (Figure 2a). Spatial quantification of biomarker expression was performed separately on the three layers of each tumor slice. Different biomarkers showed different levels of loco-regional changes. The heterogeneity of ovarian patient samples was higher than the CDX. After 3 days of culture, tumor slices from MCF-7 xenografts showed a biomarker expression gradient with loco-regional change for the ER and HIF1α expression for all the tumor slices from six different xenograft tumors in the MF system but not in the PAC system (Figure 2c). It remained the same after 7 days of cultivation (Figure 3b). The high expression of hormone receptor ER at the air-interface and the reduction of ER-positive staining in the filter region of the slices in the MF system were not observed in the PAC system after both 3 days and 7 days of culture. Detailed quantified data of the percentage difference in biomarker expression between the two outer layers of the tumor slices are shown in Appendix A. The proliferation marker Ki67 showed the same expression pattern. As can be seen in Figure 2b,d, HIF1α and γH2AX showed an inverse expression pattern to ER and Ki67 expression, accumulating at the filter interface of the slices. In contrast, in the PAC system, the expression of the biomarkers was more homogeneously distributed and closer to the in vivo situation even after 7 days of culture. The quantification data, as shown in Figure 2d, indicate that the overall ER expression of MCF-7 xenograft tumor slice cultures in the PAC system for 3 days had significantly higher expression compared to the MF culture system and were closer to the in vivo situation.

Similar results were obtained using slices from H1437 xenografts. The slices cultured in the PAC system presented similar HIF1α expression patterns to the day 0 tissues. The induced expression of Ki67 and the reduced expression of γH2AX at the air-interface of the tumor slices in the MF system was not observed in the PAC system (Figure 4a,b). All of the analyzed tumor slices from three different xenograft tumors in the MF system showed a gradient for HIF1 α, γH2AX, and CC3 (Figure 4b). In Figure 4c, IHC quantification showed a clear trend of lower Ki67 expression and higher CC3 expression after 3 days of culture in the MF system compared to the PAC.

The biomarker expression of primary human OV was highly heterogeneous between different patients. Not all of the human ovarian tumor slices showed loco-regional changes of biomarker expression with the MF culture system. Although the patient ovarian tumor slices have a high heterogeneity of biomarker expression, we can still observe the highly loco-regional changes induced in the MF culture system for some patient tumors (Figure 5b). In 9 out of 15 primary OV tumors, a gradient of biomarkers expression in the MF system was observed after 3 days cultivation, especially regarding Ki67 and HIF1α expression. The strong loco-regional changes from the air-liquid side to the filter side were not observed in the PAC system after 3 days or even after 8 days of cultivation (Figure 5). The detailed quantified data of the percentage differences are shown in Appendix A.

### 3.4. Cultivation of Tumor Slices Induces Similar Changes in Key Stress Pathways in MF and PAC System

To have a broader view of the impact of slice cultivation on tissue viability, we analyzed changes in the expression of genes that were described as being biomarkers related to cellular stress in tumor slices using Fluidigm microfluidic dynamic qPCR arrays (48.48 and 96.96 chip formats), in combination with 134 TaqMan^®^ assays according to the methods previously published [12]. The tumor slices were analyzed 3 days after culture initiation (MCF-7, H1437, and primary OV tissues) in the MF and PAC systems. The day 0 (d0) samples that had been sliced but not cultivated were also included if the in vivo samples were not available. To compare the overall impact of changes in stress-related biomarkers for the two different culture conditions, we used principal component analysis (PCA) and Euclidian distance scores (Figure 6).

In accordance with the histological observations (Figure 2a), the MCF-7 xenografts’ tumor slices cultured in the MF system or the PAC system showed similar cluster patterns to the in vivo and day 0 non-cultivated samples (Figure 6a). This was confirmed by Euclidean distance scores which represent the relationship between the cultivated slice samples and the in vivo samples (Figure 6d). The absolute number of significantly changed transcripts was slightly less in the PAC system compared to the MF system, especially with regards to the upregulated genes (Figure 7a). The fold-changes of individual biomarker transcripts are represented in Appendix A. For the H1437 xenografts tumor slices, the PAC system clustered closest to the day 0 samples (Figure 6b). This was confirmed by Euclidean distance scores (Figure 6e). The absolute number of significantly changed transcripts was lower in the PAC system compared to the MF system for both the upregulated and downregulated genes (Figure 7a). The primary OV tissues showed a high heterogeneity of cluster patterns already for the in vivo samples (Figure 6c), while the MF and PAC systems displayed similar cluster patterns. The MF system had smaller Euclidean distance scores compared to the PAC system (Figure 6f). The PAC system showed a higher absolute number of significantly changed transcripts compared to the MF system for both the upregulated and downregulated genes (Figure 7a). However, due to the high heterogeneity of the primary OV, it is hard to draw a conclusion from the data of primary OV tissues.

Classification of stress-related transcripts according to their involvement in cellular processes (Appendix A) revealed that the majority of transcripts were downregulated during slice culture with both the MF and PAC systems (Figure 7). Upregulation of transcripts was predominantly observed for apoptosis (H1437 MF system, primary OV PAC system), p38/JNK (MCF-7 MF system), and ROS (primary OV PAC system) (Figure 7). Downregulation of transcripts was predominantly observed for cell cycle (H1437 MF and PAC systems), apoptosis (MCF-7 MF and PAC systems) (Figure 7c), and cell cycle and apoptosis (primary OV in PAC system) (Figure 7d). Overall, the data indicated that the PAC and MF systems have similar impact on stress gene expression.

### 3.5. The Individual Drug Response Can Be Evaluated in the PAC System

To analyze responses to therapy, tumor slices from lung model H1437 xenografts and primary OV tumor slices were cultured in the MF and PAC systems and treated with 13 µM cisplatin, a DNA-crosslinking drug which represents a key treatment option for both lung and ovarian cancer [7,21,22]. For H1437 xenografts, cisplatin treatment led to a general enhancement of γH2Ax, Ki67, and CC3 expression in both MF and PAC systems. The gradients from the air-liquid side to the filter side of Ki67 and γH2AX expression were maintained after cisplatin treatment. The gradients were not observed in the PAC system in both the control and treated groups (Figure 8a). For primary OV slices, the expression of γH2AX and CC3 was induced after 24 h cisplatin treatment and increased further up to 72 h (Appendix A). Cisplatin treatment did not change Ki67 expression of the tumor slices in the primary OV slices (Appendix A). CC3 showed slightly higher expression in the PAC system after 3 days of treatment (Figure 8b). The different primary OV showed distinct responses to cisplatin treatment (Figure 9c). The treatment induced a similar pattern of significantly increased γH2AX expression in both MF and PAC systems (Figure 9b) while strongly enhanced CC3 was observed only in the PAC system in some patients. In Figure 9c, orange and green circles mark the corresponding data of the patients shown in Figure 9a with orange and green frames. The patient marked with orange had a strong CC3 induction after cisplatin treatment compared to the patient marked with green, although the γH2AX expression pattern did not show a big difference between these two patients. After cisplatin treatment, the overall induction of CC3 was higher in the PAC system than in the MF system (Figure 9b). This might reflect a higher aerobic metabolism in the PAC system. Overall, the individual drug response can be evaluated in the PAC system.

### 3.6. The Tumor Microenvironment of Tumor Slices Is Preserved and Treatment-Induced PD-L1 Expression Can Be Detected in the PAC System

The heterogeneous cell composition of solid tumors makes the tumor microenvironment important to identify clinically relevant drugs and patient responses to specific therapeutics. The tumor slices cultured in the PAC system have a preserved tumor microenvironment. Different cell types can be detected in the tumor slices after 3 days of culture, also after cisplatin treatment (Figure 10a). The heterogeneous cell composition observed in the in vivo tissue before cultivation is sustained. To test whether the PAC system is suitable to detect the immune response in tumor slices, we analyzed the immune microenvironment in tumor slices before and after cultivation. Using histopathological methods, the biomarkers of CD8+ for cytotoxic T cells, CD4+ for T-helper cells, CD68+ for macrophages, FOXP3+ for regulatory T cells, programmed cell death protein 1 (PD-1), and programmed cell death-ligand 1 (PD-L1) were analyzed in primary OV tumor slices for in vivo and in 3 day cultured samples. The patient-specific immune cells and their composition were preserved throughout the culture period in the PAC system after 3 days of culture and cisplatin treatment (Figure 10b). PD-1 had low expression levels in the OV tumors and did not change after tumor slice cultivation (Figure 9c,d). Patients showed an individual induction of PD-L1 after cisplatin treatment. Compared to the untreated control group, PD-L1 expression was increased after 3 days of cisplatin treatment in the PAC system for the patients with relative high PD-L1 expression (Figure 10c–e). Patients with very low PD-L1 expression did not show PD-L1 induction after cisplatin treatment. The patient with the highest PD-L1 expression level in the in vivo tumor showed the highest induction after cisplatin treatment (Figure 10e). This indicates that the tumor slices cultured in the PAC system can be used to analyze not only the individual drug response but also the immune response of the patient.

We further proved that the multiplex imaging technology can be applied to the tumor slice samples and we can observe the spatial relationship between the tumor, stromal, and immune cell components in the tumor slices. As shown in Figure 10f, after 8 days of culture, the structure of the OV tissue was preserved and the EpCAM+ tumor cells showed proliferation with Ki67 expression (Figure 10f). After cisplatin treatment, not only did EpCAM+ cells displayed PD-L1 induction but other cells did as well (Figure 10g). Except for the EpCAM+ cells, which expressed γH2AX and CC3, some CD3+ cells also showed positive expression of γH2AX and CC3. This indicates that the T cells in the tumor microenvironment may also react to cisplatin treatment (Figure 10g). Further analysis should be carried out with more samples to draw clearer conclusions about it.

## 4. Discussion

Precision-cut tumor slice culture has the ability to closely recapitulate the architecture and heterogeneity of the original tumors. Therefore, it can be used as a platform to study the tumor microenvironment and evaluate the preclinical efficacy of drug treatment. As a model, the tumor slice culture also has limitations that cannot be overcome with the currently available culture methods. A major limitation is the lack of functional vasculature in the tumor slice [14]. Due to the loss of the intact circulatory network in the tumor slice, the availability of oxygen, nutrients, or drugs in tumor slices is strictly limited to diffusion [12,23]. In in vivo tumors, oxygen is provided by hemoglobin in red blood cells. Hemoglobin can deliver large amounts of oxygen to cells at relatively low oxygen tension. This unique property simply cannot be replicated in vitro [24]. In the cell culture medium, an efficient oxygen carrier such as hemoglobin does not exist. It was shown that the oxygen concentration in the medium is about 2% of the oxygen content in the ambient atmosphere [25]. The total amount of oxygen that can be supplied from the medium to the tumor slice is very limited. Studies have tried to increase the amount of dissolved oxygen in culture medium by using up to 95% oxygen in the culture. However, the increase in the oxygen concentration is still limited and may induce hyperoxia problems in the tissue [24,26]. Because of hemoglobin, blood as a liquid has the ability to offer comparable amounts of oxygen as the atmosphere, which has approximately 10 times higher oxygen content (delta of artery and venous blood oxygen content) than the medium (saturated with atmospheric oxygen) even under lower oxygen partial pressure with 100 mmHg [27,28]. Therefore, in vivo, the oxygen can be quickly transported and rapidly released to the tissue even in a low oxygen partial pressure environment. The special ability of hemoglobin ensures sufficient oxygen supply to the tissue without the problem of hyperoxia. The tumor slices do not have a functional blood supply; the oxygen must be diffused from ambient gases into the medium and then into the tumor slices. The PAC system has two sides of air-liquid interfaces, which can minimize the diffusion distance from the ambient gases to the tumor slice; therefore, it can provide a sufficient oxygen supply to the tumor slices during cultivation under a low oxygen tension. For the experiments, a normal tumor slice with 200–300 µm thickness and 5–10 mm diameter contains more than 5 million cells, which is about the same cell number of a confluent T75 cell culture flask. Normally, up to 20 mL medium is used for the 2D monolayer cell culture with the T75 flask. The commonly used Millipore filter (MF) system for tumor slice culture can only apply about 1.5 mL medium under static conditions, which is much less than the 2D monolayer cell culture system. This also explains the gradients from the air to the filter interface of the tumor slice cultured in the MF system. The tumor slice cultivated in the MF system has two different sides, the air-interface side and filter-liquid side. In the in vivo tumor, the oxygen, nutrients, or drugs can diffuse from the blood vessels to the cells [1]. In the MF system, they are mainly provided from two different directions to the tumor slices. The oxygen can only diffuse mainly from the air-liquid interface side of the tumor slice, and the nutrients and drugs are mostly supplied from the filter-liquid side and have to first pass through the filter to reach the tumor slices. Considering that the MF system is a static culture, the exchange of nutrients has low efficiency. It cannot faithfully recapitulate the in vivo situation. The development of loco-regional heterogeneity in the MF culture system has been reported for both xenografts and patient tumors [12]. This can also be observed in our data. Although gradients of oxygen tension are a common feature of solid tumors [29], prudence is required when interpreting the data from the MF system culture with the gradients, especially after drug treatment, because the oxygen, nutrients, or drugs may be supplied from two separate sides of the tumor slice.

The PAC system overcomes this problem. The oxygen, nutrients, and drugs are supplied from both sides of the tumor slices with the same direction, which is closer to the in vivo situation. The cotton meshes on both sides of the tumor slice work not only as a support structure but also as vasculature in tumor slices. With this special structure, the thickness of the tumor slice can be increased and is not limited to 200–300 µm for cultivation. The hypoxic condition can also be created in the middle of the tumor slices when changing the oxygen supply to hypoxia in the PAC system. As one of the key parameters governing tissue viability, organotypic supports must fulfill two functions. One is to mimic the vasculature and supply sufficient oxygen and nutrients to the tumor tissue continuously; the other is to act as the extracellular matrix (ECM) to protect the thin and friable slice from the stress of fluidic flow and keep the stiffness of the tumor slices. It has been well known that tumors are much stiffer than normal tissues because of the changes in their ECM. The relative stiffness can have profound effects on cellular function [1]. We have compared many different materials as organotypic supports for the PAC system. Nylon meshes with different pore sizes (89 µm, 41 µm, and 29 µm), polycarbonate membranes with a 12 µm pore size, and hydrogels did not show good oxygen and nutrients supply because of their small pore sizes and material structures (data not shown). Cotton meshes with 500 µm pore size showed the best oxygen, nutrient, and drug supply. Because of their big pore size, some cells on the surface of the slices were washed away during the culture in the PAC system. Therefore, we mainly used cotton meshes as organotypic supports for 3-day, short-term experiments. The decellularized scaffold from the porcine intestine can better protect the slices from the shear stress of fluidic flow. We used it together with cotton meshes for long-term cultivation of the tumor slices. Thus, the primary OV tumor slices can be cultured for up to 8 days within the PAC system. Although the cotton mesh showed better properties than the other organotypic supports tested, it still has drawbacks. It may wash away cells from tumor slices during culture and it is difficult to section the cotton fibers together with the tissue in the formalin-fixed paraffin-embedded (FFPE) samples for IHC staining. The scaffold from the porcine intestine does not have this problem, but it still requires the addition of a cotton mesh to maintain a better distribution of the medium to the slices. In addition, it was presumed that atmospheric concentrations of oxygen can produce hyperoxic conditions at the air-liquid interface side to allow a supply of just sufficient oxygen to deeper cell layers [12]. In the PAC system, if the oxygen can be supplied from both sides of the tumor slice, the oxygen concentration in the atmosphere can be reduced to avoid hyperoxic damage to the tumor slices. Overall, the PAC system still needs to be further optimized especially with regards to supports and a suitable oxygen concentration surrounding the tumor slices during culture.

Cultivation of tumor slices induces similar changes in key stress pathways in the MF and PAC systems. This result can be confirmed with the Euclidean distance of stress gene biomarker expression. The in vivo primary OV samples showed a high heterogeneity of the cluster pattern in the PCA plots, making it hard to draw a conclusion from the primary OV tissue data. Therefore, IHC staining is a better method to evaluate the primary OV tumor slices. Because of the highly inter- and intratumoral heterogeneity of the patient tumors, the tissue structure and the spatial distribution of biomarker expression in tumors are more important.

Unlike the CDX tumor slices, the primary OV tumor slices showed gradient patterns in the MF system culture in only 9 out of 15 cases. There could be two reasons for this. One is the heterogeneity of the primary OV of patients; the other is that the proliferation rate of the tumor cells in primary OV is much slower than the tumor cells in CDX. This can also explain why the cisplatin treatment induced Ki67 expression in H1437 CDX but not in primary OV tumor slices. Different primary OV tumors showed different responses to cisplatin treatment. This also reflects the heterogeneity of the primary OV and indicates the necessity of personalized therapy. Cisplatin treatment was accompanied by a minor increase in γH2AX in both MF and PAC systems while strongly enhanced CC3 was observed only in the PAC system. This might reflect higher aerobic metabolism in the PAC system [22]. This indicates that the functional response to drug treatment is more sensitive in the PAC system. Further studies will need to be undertaken which correlate the responses of primary OV tumor slices to cisplatin treatment with clinical outcomes to predict patient response, especially the large difference in CC3 induction between primary OV tumor slices.

Several reports have shown that tumor slices can be used to study the immune microenvironment and test the immune responses of drug treatment [10,18]. Here, we provide evidence that the tumor slices cultured in the PAC system are suitable for studying the tumor immune cell environment and the immune response of drug treatment. Interestingly, PD-L1 expression was increased individually after cisplatin treatment compared to the untreated control group in primary OV tumors (Figure 10d,e). The PAC system showed comparable and even more sensitive data in comparison to the MF system (Appendix A). Other studies have shown that non-small cell lung cancer patients who received cisplatin-based neoadjuvant chemotherapy followed by surgery have significantly increased PD-L1 expression in both tumor cells and immune cells from the microenvironment [30]. Our data also demonstrated that not only the EpCAM+ cells showed increased PD-L1 expression. It is also reported that cisplatin induces PD-L1 over-expression in hepatoma H22 cells [31]. Cisplatin also upregulates PD-L1 expression in vitro and in vivo in ovarian cancer mouse models [32]. Our results on tumor slices support the evidence from previous observations and indicate that the PAC system is a reliable system for studying drug treatment and predicting in vivo drug response.

## 5. Conclusions

Cultivation of precision-cut tumor slices in the novel PAC system provides a preclinical model that preserves tumor heterogeneity and the native microenvironment. As the PAC system facilitates the homogenous and controlled supply of oxygen, nutrients, and drugs, it is suitable for assessing individual drug responses and can thus be used as a preclinical model to predict in vivo therapy response.

## 6. Patents

Dong M, Schwab M, and van der Kuip H (WO/2019/029947; EP 20180742758) device for cultivating tissue sections.

## Figures and Tables

**Figure 1 cells-12-00807-f001:**
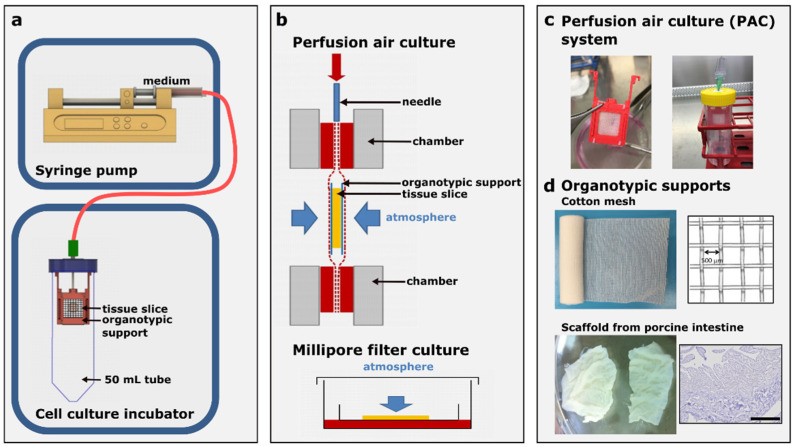
Details of the perfusion air culture (PAC) system for tumor slice culture. (**a**) The precision-cut tumor slices (250 µm to 280 µm thickness) are cultured in between two organotypic supports and fixed in a special chamber allowing continuous perfusion with the medium and drugs (perfusion rate: 2 mL per day). The chamber is settled vertically inside of a 50 mL tube with air exchange capacity and connected to a syringe pump via a silicone tube. The system is placed inside a cell culture incubator. (**b**) Cross-section of the setup of the PAC system and the commonly used Millipore filter (Millicell Cell Culture inserts) culture system. (**c**) An actual sample of the chamber packed with the tumor slice and the chamber inside the PAC system for cultivation. (**d**) Two different types of organotypic supports were used in the PAC system for different cultivation purposes: cotton meshes with a 500 µm opening size and scaffolds from a porcine intestine as shown by the H&E-stained structure. The scale bar represents 100 µm.

**Figure 2 cells-12-00807-f002:**
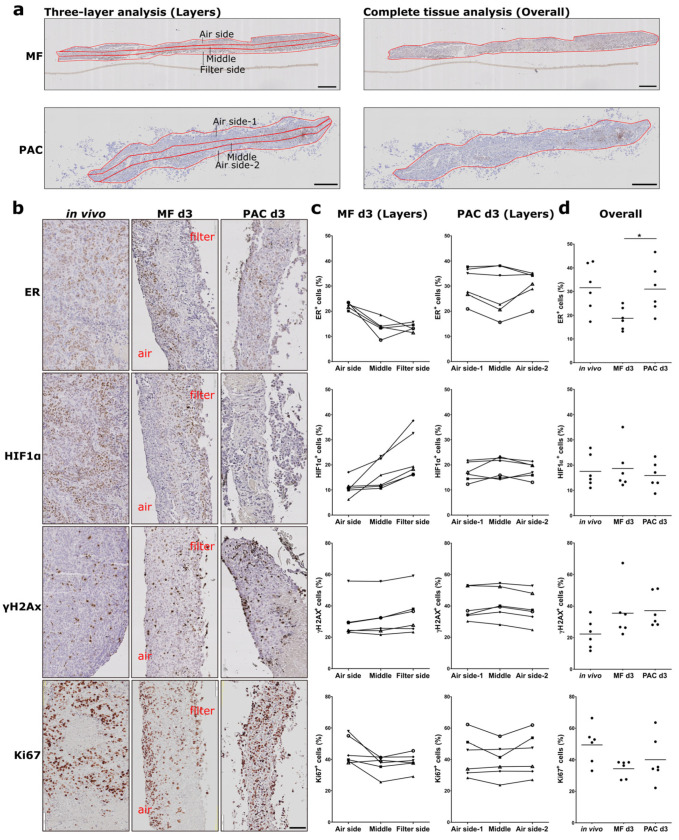
Expression of estrogen receptor (ER), hypoxia-inducible factor 1α (HIF1α), phospho-histone H2A.X (γH2AX), and Ki67 of MCF-7 cell-line-derived xenograft (CDX) tumor slices in the Millipore filter (MF) culture system compared to the perfusion air culture (PAC) system after 3 days of culture. (**a**) Biomarker expression was determined either in different longitudinal layers of the tumor slices (MF: Air side, Middle, Filter side; PAC: Air side-1, Middle, Air side-2) (Layers) or across the complete tissue (Overall). (**b**) Immunohistochemical (IHC) staining of biomarkers in MCF-7 CDX tumor slices after 3 days of cultivation either statically on a Millipore filter (MF) or in the perfusion air culture (PAC) system and compared to the original in vivo tumor. Air indicates the air side; filter indicates the filter side of the MF culture system. The scale bar represents 100 µm. (**c**) Quantification data of the percentage of cells expressing ER, HIF1α, γH2AX, and Ki67 in different layers of the tumor slices as defined as (**a**) after culture in the MF or PAC systems from six mice. In the figure, each shape of the symbol represents one mouse experiment. (**d**) Quantification data of the overall percentage of cells in whole tumor slices expressing ER, HIF1α, γH2AX, and Ki67 in the MF and PAC systems and in vivo tumors from six mice (* *p*-value < 0.05).

**Figure 3 cells-12-00807-f003:**
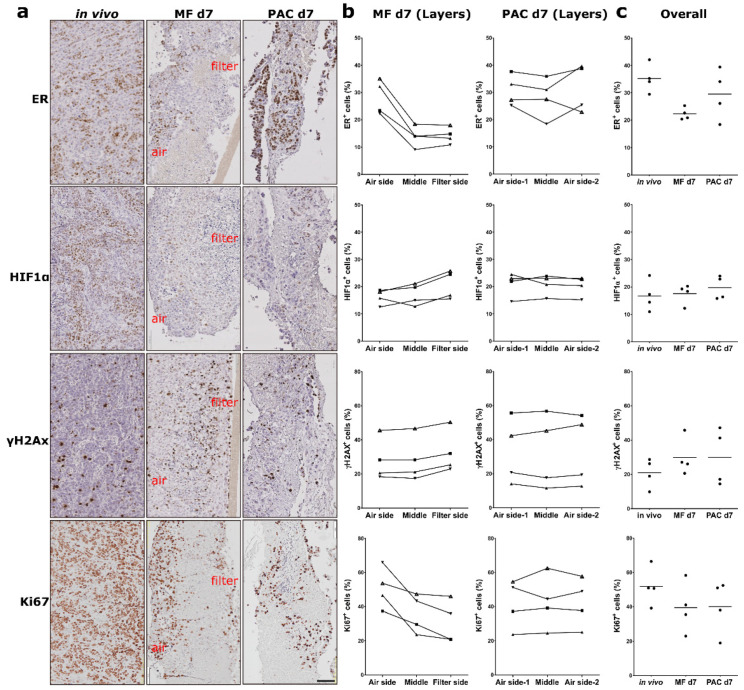
Expression of estrogen receptor (ER), hypoxia-inducible factor 1α (HIF1α), phospho-histone H2A.X (γH2AX), and Ki67 of MCF-7 cell-line-derived xenograft (CDX) tumor slices in the Millipore filter (MF) culture system compared to the perfusion air culture (PAC) system after 7 days of culture. (**a**) Immunohistochemical (IHC) staining of biomarkers in MCF-7 CDX tumor slices after 7 days of cultivation either statically on a Millipore filter (MF) or in the perfusion air culture (PAC) system and compared to the original in vivo tumor. Air indicates the air side; filter indicates the filter side of the MF culture system. The scale bar represents 100 µm. (**b**) Quantification data of the percentage of cells expressing ER, HIF1α, γH2AX, and Ki67 in different longitudinal layers of the tumor slices after MF or PAC culture from four mice. In the figure, each shape of the symbol represents one mouse experiment. (**c**) Quantification data of the overall percentage of cells in whole tumor slices expressing ER, HIF1α, γH2AX, and Ki67 on MF, PAC systems and in vivo tumors from four mice.

**Figure 4 cells-12-00807-f004:**
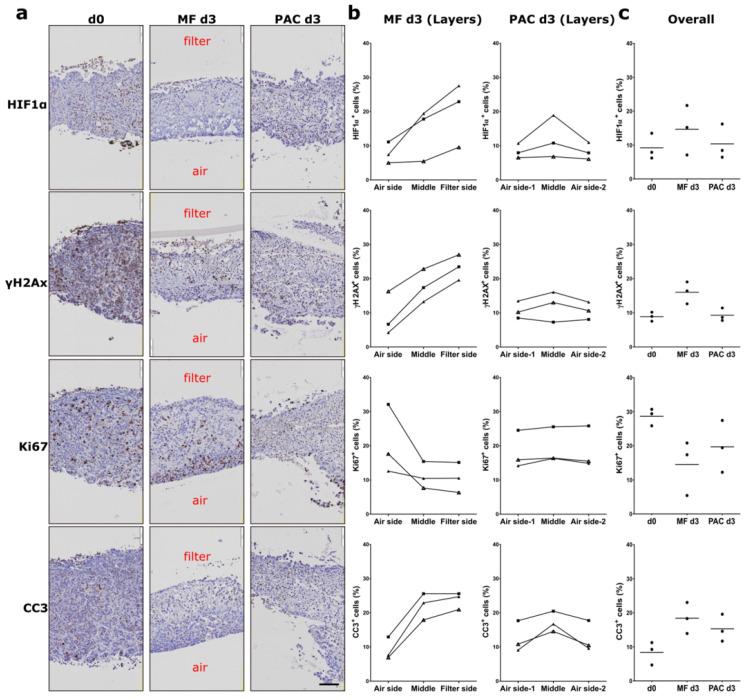
Expression of hypoxia-inducible factor 1α (HIF1α), phospho-histone H2A.X (γH2AX), Ki67, and cleaved-caspase 3 (CC3) of H1437 CDX tumor slices in the Millipore filter (MF) culture system compared to the perfusion air culture (PAC) system. (**a**) Immunohistochemical (IHC) staining of biomarkers in H1437 CDX tumor slices after 3 days of cultivation either statically on Millipore filter (MF) or in the perfusion air culture (PAC) system with cotton meshes as organotypic support and compared to the day 0 (d0) non-cultivated tumor slices. Air indicates the air side; filter indicates the filter side of the MF culture system. Ki67, HIF1α, and γH2AX were stained to investigate proliferation, oxygen supply, and DNA damage. The scale bar represents 100 µm. (**b**) Quantification data of the percentage of cells expressing HIF1α, γH2AX, Ki67, or CC3 in different longitudinal layers of the tumor slices after culture in the MF or PAC systems. In the figures, each shape of the symbol represents one mouse experiment. Data shown are from experiments of three mice. (**c**) Quantification data of the overall percentage of cells in whole tumor slices expressing HIF1α, γH2AX, Ki67, and CC3 after culture in MF or PAC systems or in the in vivo tumors. Data shown are from experiments of three mice.

**Figure 5 cells-12-00807-f005:**
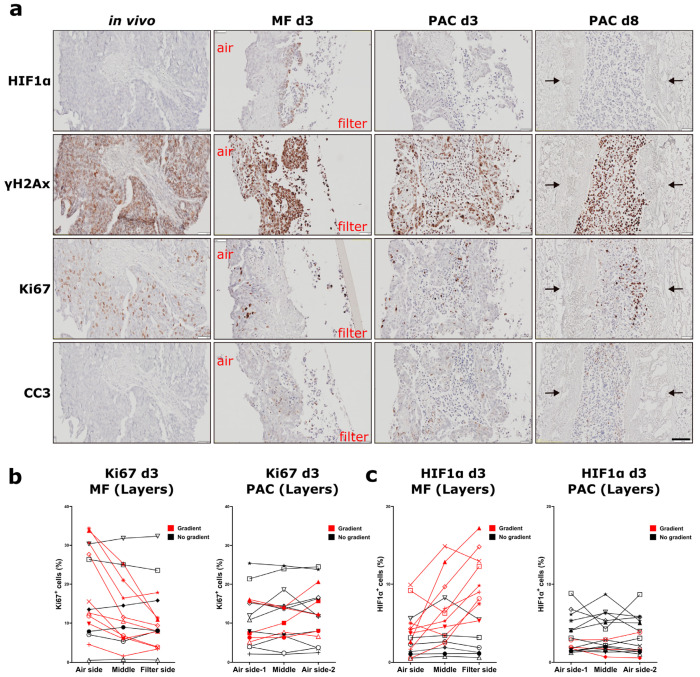
Expression of hypoxia-inducible factor 1α (HIF1α), phospho-histone H2A.X (γH2AX), Ki67, and cleaved-caspase 3 (CC3) of primary human ovarian tumor slices in the Millipore filter (MF) culture system (3 days) and perfusion air culture (PAC) system (3 days and 8 days). (**a**) Immunohistochemical (IHC) staining of biomarkers in primary human ovarian tumor slices after 3 days cultivation statically in the Millipore filter (MF) system and in the perfusion air culture (PAC) system with cotton meshes as organotypic support, or after 8 days in the PAC system with a scaffold from a porcine intestine as organotypic support and compared to the in vivo tumors. Air indicates the air side; filter indicates the filter side of the MF culture system. HIF1α, γH2AX, Ki67, and CC3 were stained to investigate oxygen supply, DNA damage, proliferation, and apoptosis. After 3 days of culture, the tumor slices cultured in the static MF culture system showed induction of HIF1α and γH2AX expression at the filter side. Conversely, Ki67-positive cells were found at the air side of the tumor slices. The tumor slices cultured in the PAC system showed similar morphology and biomarker expression to the in vivo tumor tissue after both 3 days (d3) and 8 days (d8) of culture. Arrows indicate the scaffold from the porcine intestine. The tumor slices were embedded vertically together with organotypic supports in the FFPE blocks. The Millipore filter, cotton, and scaffold (indicated with arrows) can be observed in the IHC images in the figures. The scale bar represents 100 µm. (**b**) Quantification data of the percentage of positive cells expressing Ki67 in different longitudinal layers of the tumor slices after 3 days of MF or PAC culture. In the figure, each shape of the symbol represents an individual patient experiment. Data shown are from experiments of 15 patients. (**c**) Quantification data of the percentage of cells expressing HIF1α in different areas of the tumor slices after 3 days of MF or PAC culture. In the figure, each shape of the symbol represents an individual patient experiment. Data shown are from experiments of 15 patients.

**Figure 6 cells-12-00807-f006:**
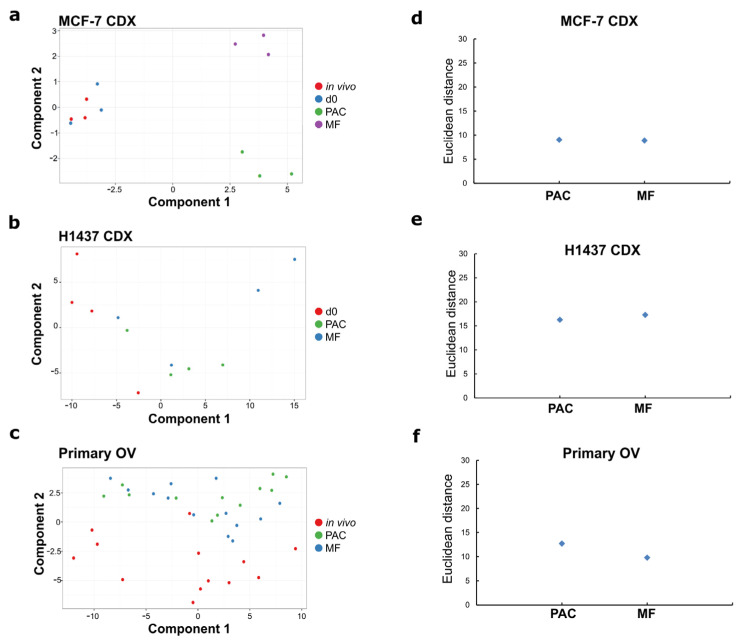
Evaluation of the changes in stress biomarkers under different conditions of tumor slice culture by principal component analysis (PCA). (**a**–**c**) PCA plots of the expression of 134 stress genes quantified under different conditions of tumor slice culture. (**d**–**f**) Scatterplots of the Euclidean distance of stress gene biomarker expression profiles of tumor slices compared to the expression profiles of the corresponding in vivo or d0 tumors. The tissue fixed immediately after surgical resection was defined as an in vivo tumor. The tissue after the slicing process and before tumor slice cultivation was defined as a d0 tumor. The results are from three mice of MCF-7 CDX for figures a and d, from four mice of H1437 CDX for figure b and e, and from 13 patients with ovarian tumors for figures (**c**,**f**).

**Figure 7 cells-12-00807-f007:**
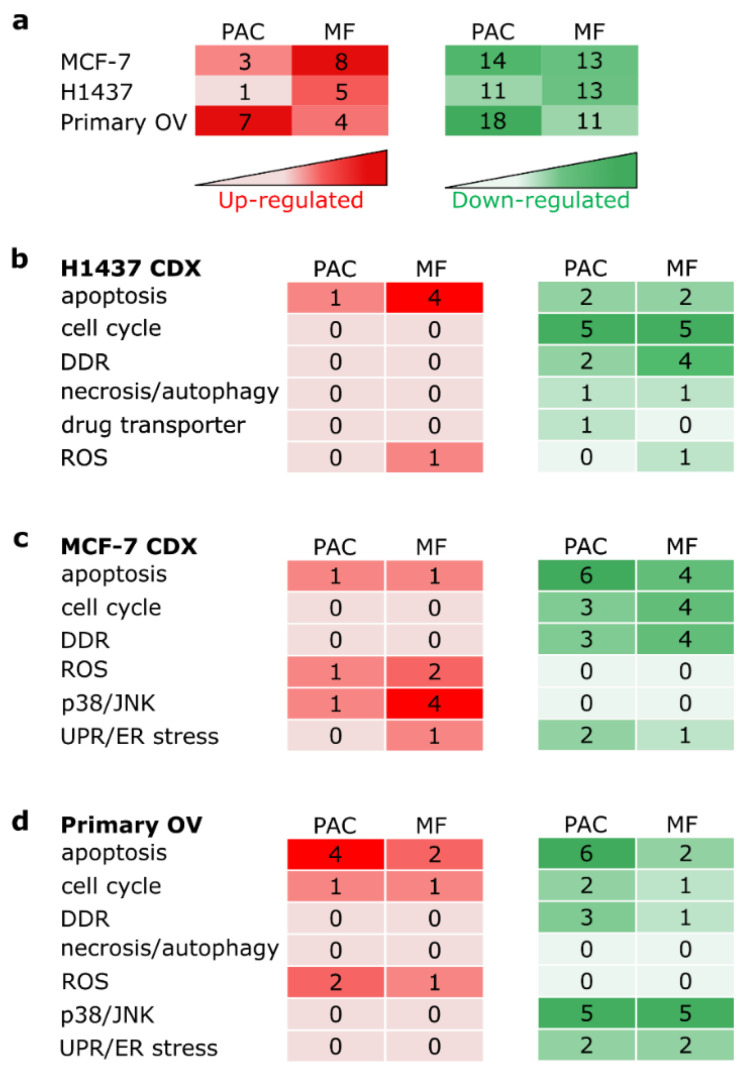
Pathway changes induced by tumor slice culture under different conditions. (**a**) Table showing the numbers of differentially expressed genes (DEGs) in breast model MCF-7 CDX, lung model H1437 CDX, and primary human ovarian tumor tissue (primary OV) under Millipore filter (MF) and perfusion air culture (PAC) culture conditions. (**b**–**d**) Tables showing the numbers of DEGs under MF and PAC cultivation conditions and their associated functions for breast model MCF-7 CDX, lung model H1437 CDX, and primary human ovarian tumor tissue (primary OV). The results are from 3 mice of MCF-7 CDX, from 4 mice of H1437 CDX, and from 13 patients with ovarian tumor.

**Figure 8 cells-12-00807-f008:**
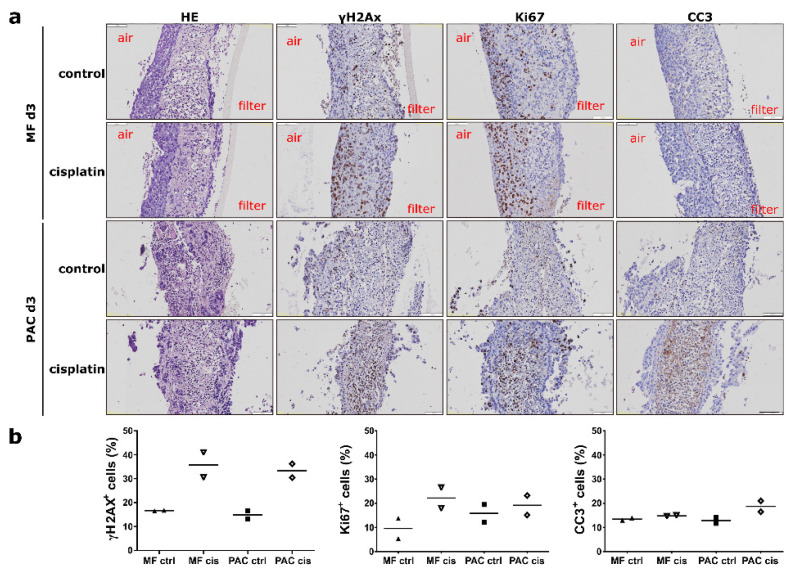
Cisplatin treatment of H1437 CDX tumor slices. (**a**) The tumor slices were incubated with cisplatin for 3 days in both the Millipore filter (MF) system and the perfusion air culture (PAC) system. Compared to the untreated control group (control), the cisplatin-treated tumor slices (cisplatin) cultured in the MF showed the accumulation of γH2AX and Ki67 at the air interface. The expression of cleaved-caspase 3 (CC3) was not observed after cisplatin treatment in the MF system. After cisplatin treatment, CC3, γH2AX, and Ki67 were induced in the middle of the tumor slices in the PAC system. The effects of cisplatin in the tumor slices were higher in the PAC system compared to the MF system. The scale bar represents 100 µm. (**b**) Quantification data of the percentage of positive cells expressing γH2AX, Ki67, and CC3 in the MF and PAC systems with cisplatin treatment from two mice experiments. In the figures, each shape of the symbol represents one mouse experiment.

**Figure 9 cells-12-00807-f009:**
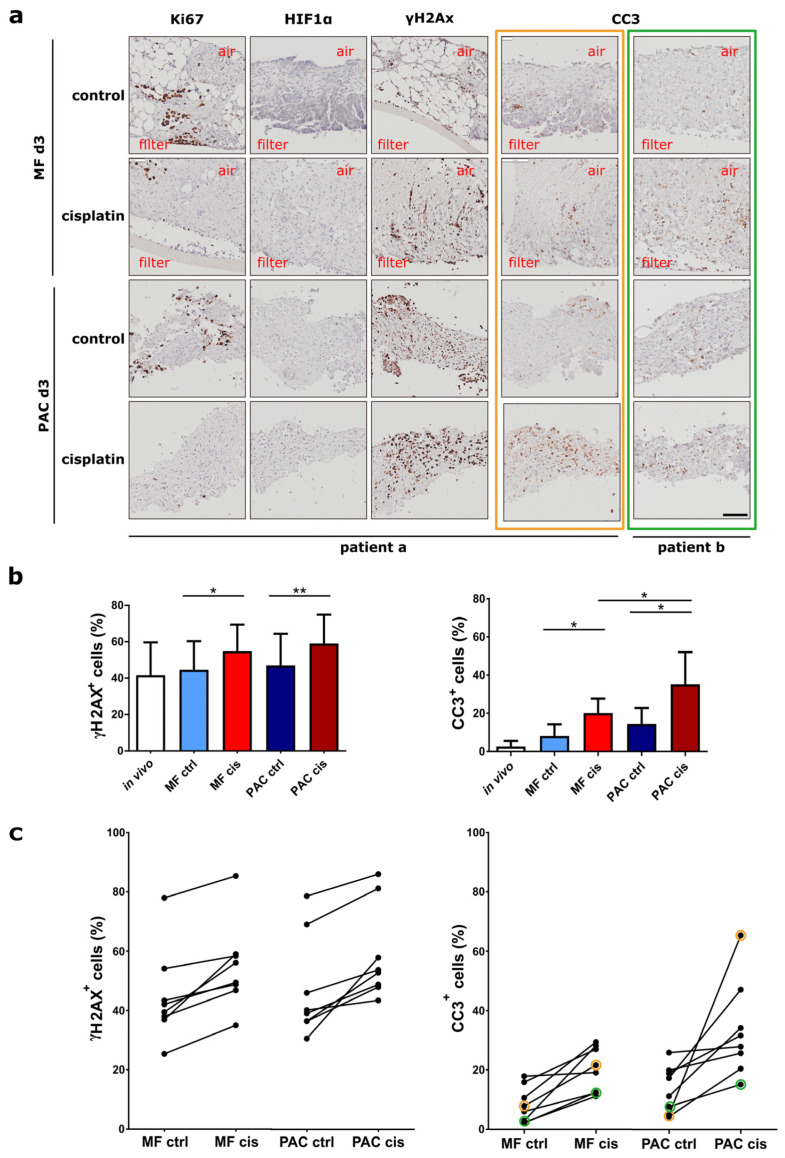
Treatment of cisplatin on primary ovarian tumor slices. (**a**) The primary ovarian tumor slices were incubated with cisplatin for 3 days in both the Millipore filter (MF) system or the perfusion air culture (PAC) system. Cisplatin treatment did not influence Ki67 and HIF1α expression in primary ovarian tumor slices. A minor increase in γH2AX was observed in both MF and PAC systems. Different patient tumors (marked with orange and green with frames) showed different CC3 expression, while only in the PAC system was strongly enhanced CC3 observed. The scale bar represents 100 µm. (**b**) The overall percentage of positive cells expressing γH2AX and CC3 on MF and PAC systems after 3 days culture without (ctrl) and with cisplatin (cis) treatment from eight patient experiments. (**c**) The percentage of positive cells expressing γH2AX and CC3 after 3 days culture for each patient without (ctrl) and with cisplatin (cis) treatment (n = 8) in the Millipore filter (MF) system or the perfusion air culture (PAC) system. The two black dots connected by a straight line in the figure are from the experiment of one patient. The orange and green circles in CC3 expression indicate the corresponding patients in Figure (**a**) marked with orange and green frames. (* *p*-value < 0.05, ** *p*-value < 0.01).

**Figure 10 cells-12-00807-f010:**
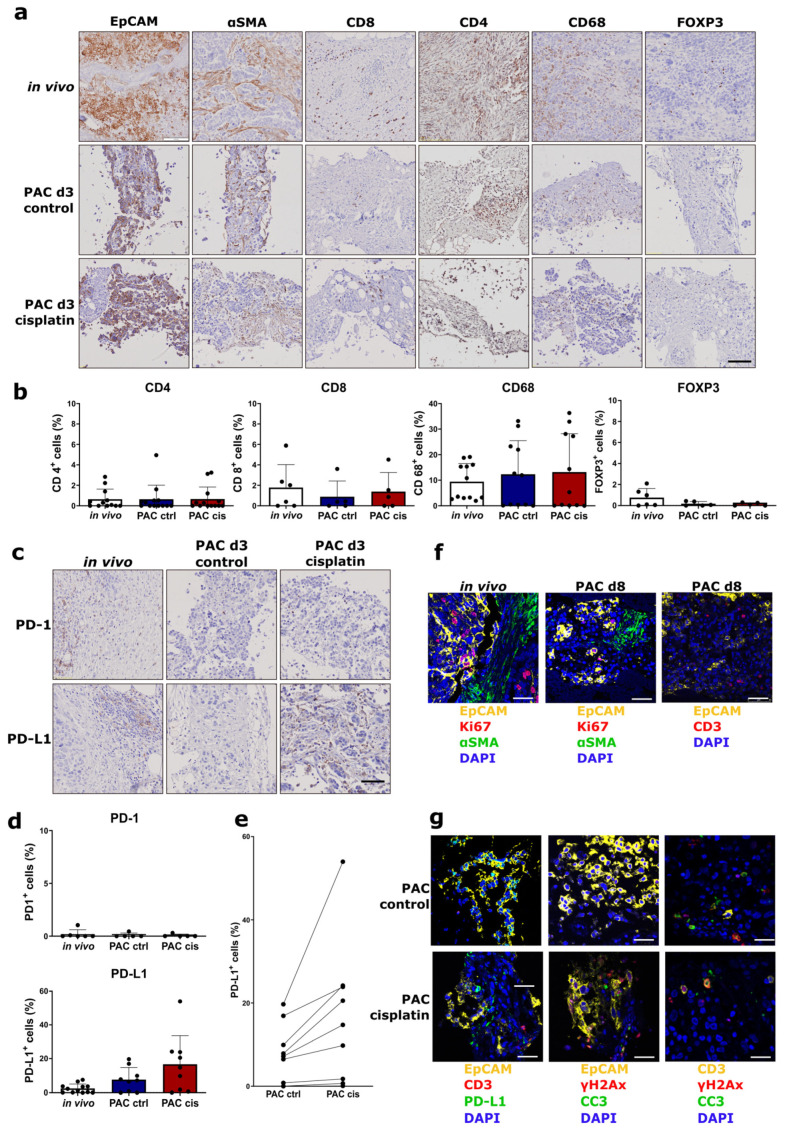
The tumor microenvironment including immune cells are preserved in tumor slice culture in the PAC system. The primary human ovarian tumor slices were cultured in the perfusion air culture (PAC) system without (ctrl) or with cisplatin (cis) treatment for 3 days. (**a**) Immunohistochemical (IHC) staining of biomarkers expression for tumor cells (EpCAM), fibroblasts (aSMA), T cells (CD4, CD8, and FOXP3), and macrophages (CD68) in tumor slices compared to the in vivo tumors. The scale bar represents 100 µm. (**b**) Percentage of CD4-, CD8-, CD68-, and FOXP3-positive cells in tumor slices and in vivo tumors from different patients (n = 12 or 6). In the figures, each black dot represents one patient tumor. (**c**) IHC staining of key proteins of the checkpoint inhibition system PD-1 and PD-L1 in tumor slices compared to the in vivo tumors. The scale bar represents 100 µm. (**d**) Percentage of PD-1- and PD-L1-positive cells in tumor slices without (ctrl) or with cisplatin (cis) treatment compared to the in vivo tumors from different patients (n = 9 or 6). In the figures, each black dot represents one patient tumor. (**e**) Cisplatin treatment induced PD-L1 expression after 3 days of culture in tumor slices from different patients (n = 9). The two black dots connected by a straight line in the figure are from the experiment of one patient. (**f**) Multiplex staining for different cell types in the in vivo tumor and tumor slices after 8 days of culture in the PAC system (EpCAM, Ki67, aSMA, CD3, and DAPI). (**g**) Multiplex staining of control and cisplatin-treated tumor slices for different cell types and γH2AX, CC3, and PD-L1 expression. The scale bar represents 50 µm.

## Data Availability

Original data supporting the findings of this study are available upon request from the corresponding authors.

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
