# Peer review of "Perfusion Air Culture of Precision-Cut Tumor Slices: An Ex Vivo System to Evaluate Individual Drug Response under Controlled Culture Conditions"

_cells, 2023, doi:10.3390/cells12050807_

Round 1
Reviewer 1 Report
The paper by Dong et al describes an interesting methodology to preserve viability of tumor slices for studies of the tumor microenvironment (TME). This is an improvement of an approach previously published (Davies et al, Sci Rep 2015). The novelty is the addition of a perfusion that provides a continuous and supply of oxygen, nutrients and drugs. The concept is simple but efficient and useful, and it was applied to mouse xenografts of two cell lines (breast MCF-7 and lung H1437) and to primary human ovarian tumors (OV). Ex-vivo drug response to a chemotherapy (cisplatin) was investigated for H1437 xenografts and human ovarian tumors.
The work present strong weaknesses in result presentation and interpretation that need to be absolutely addressed prior potential reconsideration for publication.
The method helps to preserve viability and in-vivo features of tumor slices, but cannot provide precise control of culture conditions. This is particularly true for the oxygen. There is absolutely no control of the oxygen dissolved in the medium. We recommend to remove the terms “precisely controlled” from the manuscript text, including the title and the abstract. We also ask the authors to clarify their discussion on oxygen, lines 532-549. It does not seem relevant to their results, not fully correct, since it is known that tumors do not have normal vasculature and therefore are subjected to hypoxia.
The authors claim that the major advantage of the improved method is the absence of intra-slice gradients, as assessed by looking at various marker HIF1a, gH2AX, Ki67 and CC3. Even though the presented images and quantifications suggest that this is probably true, the analyses must be completed and presented more frankly. The point is the artificial spatial heterogeneity, therefore for all the IHC experiments all the markers should be analyzed spatially, as done for example in Supplementary 2b (but filter, middle and air side must be objectively defined).
The quantifications reported in Figure 2b, 2d, 3b do not support the conclusion that “Loco-regional changes in biomarkers induced in the filter support culture system were overcome with the perfusion air culture (PAC) system”. I understand that the low number and high heterogeneity of samples may result in not-statistically significant differences. Qualitative trends, especially if reproducible, may be acceptable. I recommend to faithfully report all the spatial quantifications for each tumor slices, n=4 for MCF-7, n=3 for H1437, n=15 for OV. Indeed, it is interesting to document the patient heterogeneity of OV samples.
In the transcriptomic analysis, “Results are from 3 independent experiments for MCF-7 CDX, from 4 independent experiments for H1437 CDX, from 13 independent experiments for primary OV”. What does it mean? Are the fold changes of Supplementary Table1 the means of the experiments? If the OV data are very heterogeneous, they should not be pooled together. Please, clarify this point.
Page 11. “The primary OV tissues showed a high heterogeneity of cluster patterns already for in vivo samples (Figure 5c), while the MF and PAC systems displayed similar cluster patterns to the in vivo samples”. From what I see, the red points (in vivo) are clearly clustered separately from blue (MF) and green (PAC) points. Therefore, the conclusion does not seem to be correct, but I am not expert in PCA analysis.
About the drug response part, I agree with the conclusion that “The individual drug response can be evaluated in the PAC system”, but the presented data do not support the statements that “the functional response to drug treatment is more sensitive in the PAC system”, nor that “The effects of cisplatin on the tumor slices were higher in the PAC system compared to the MF system”.
Figure 9f and 9g only indicate that multiplex staining can be performed on PAC system. The conclusions of lines 516-522 are not supported by the provided images, maybe because of the low quality of PDF file. Quantifications of double/triple positive cells might help. Untreated controls for Figure 9g are missing.
Manuscript should be revised to improve clarity. Important information is found in the legends instead than in the main text. For example, the sentence “the HIF1α, γH2AX, Ki67 and CC3 were stained to investigate oxygen supply, DNA damage, proliferation and apoptosis” explains the rational of the experiments and is found only in the legend of Figure 4. The choice of cisplatin is clinically appropriate since this chemotherapy is used for lung and ovarian cancers, but this is never explained. Moreover, only in legend of Figure 6 it is mentioned that H1437 are used as lung cancer model.
Did the authors verify the absence of mycoplasma contaminations? Did the authors perform STR validation of cell lines?
Since this is a methodology paper, it will be important to include more precise information about material and reagents, i.e. both supplier and reference number, for example for filter papers, Matrigel, ATCC and Miltenyi references, media…
Author Response
Response to Reviewer 1 Comments
The paper by Dong et al describes an interesting methodology to preserve viability of tumor slices for studies of the tumor microenvironment (TME). This is an improvement of an approach previously published (Davies et al, Sci Rep 2015). The novelty is the addition of a perfusion that provides a continuous and supply of oxygen, nutrients and drugs. The concept is simple but efficient and useful, and it was applied to mouse xenografts of two cell lines (breast MCF-7 and lung H1437) and to primary human ovarian tumors (OV). Ex-vivo drug response to a chemotherapy (cisplatin) was investigated for H1437 xenografts and human ovarian tumors.
The work present strong weaknesses in result presentation and interpretation that need to be absolutely addressed prior potential reconsideration for publication.
We would like to thank this reviewer for the insightful analysis of our manuscript.
The method helps to preserve viability and in-vivo features of tumor slices, but cannot provide precise control of culture conditions. This is particularly true for the oxygen. There is absolutely no control of the oxygen dissolved in the medium. We recommend to remove the terms “precisely controlled” from the manuscript text, including the title and the abstract. We also ask the authors to clarify their discussion on oxygen, lines 532-549. It does not seem relevant to their results, not fully correct, since it is known that tumors do not have normal vasculature and therefore are subjected to hypoxia.
Response: We thank the reviewer for this valuable suggestion. Actually, the oxygen dissolved in the medium was controlled under a fixed condition in our culture system. We apologize for the lack of clarity of this point in the text. The detailed information was added in Materials and Methods / perfusion air culture system construction and in the results part 3.1. To be more accurate, we have removed the “precisely” from the manuscript text, including the title and the abstract.
In the PAC system, the cultivation was performed in a normal cell culture incubator at 37°C and 5% CO2 under atmospheric oxygen (21% oxygen) conditions. The cell culture medium was saturated with 21% oxygen in the incubator before it reached the tissue slice. The silicone tube with an inside diameter (ID) of 0.5mm and outside diameter (OD) of 2.5mm has a high gas permeability for oxygen. We used 1 m long silicone tube to transfer the medium through the pump to the tissue slices for each sample. With a 83µL/h (2ml/day) low perfusion rate, the cell culture medium can be well saturated with 21% oxygen condition at 37°C in the cell culture incubator. The oxygen supply for each sample was maintained at 21% oxygen independent of samples and experiments. We can also adjust the oxygen concentration in the culture medium through adjusting the oxygen percentage (from 21% oxygen to 5% oxygen or to 30% oxygen) in the cell culture incubator.
The tumors are mostly under hypoxic conditions in vivo, but this hypoxia is a dynamic situation. In vivo, the oxygen released from hemoglobin is diffused from the vascular to the tumor area and the oxygen is rapidly used by the cells in the tumor. Although the tumor has abnormal tumor blood flow and is mostly under hypoxic condition, they still use oxygen as much as they can receive from the environment. A small tumor slice with 250 µm thickness 10 mm diameter contains already more than 5 million cells. Considering the huge cell numbers of solid tumors, the tumor cells requires a large amount of oxygen even under hypoxic condition. Hemoglobin can deliver large amounts of oxygen to cells at relatively low oxygen tension. This unique property simply cannot be replicated in vitro. The discussion on oxygen (lines 532-549) was mainly to explain why the oxygen content in the atmosphere is comparable to that in arterial blood and further described the importance of the air-liquid interface to tumor slices culture from the quantitative level. The tissue slices do not have functional blood supply; the oxygen must be diffused from ambient gases into the medium then into the tumor slices. The PAC system has two sides of air-liquid interfaces, which can minimize the diffusion distance from the ambient gases to the tissue slice; therefore, it can provide sufficient oxygen supply to the tumor slices during cultivation under a low oxygen tension.
We have modified the discussion on oxygen in lines 532-549 to make it clearer and easier to understand.
The authors claim that the major advantage of the improved method is the absence of intra-slice gradients, as assessed by looking at various marker HIF1a, gH2AX, Ki67 and CC3. Even though the presented images and quantifications suggest that this is probably true, the analyses must be completed and presented more frankly. The point is the artificial spatial heterogeneity, therefore for all the IHC experiments all the markers should be analyzed spatially, as done for example in Supplementary 2b (but filter, middle and air side must be objectively defined).
Response: We thank the reviewer for this valuable feedback. We have performed the spatial analysis for the IHC images. The tumor slices were objectively divided into three layers longitudinally according to their shapes (MF: Air side, Middle, Filter side; PAC: Air side-1, Middle, Air side-2) using the QuPath software. All three layers from each tumor slice were analyzed separately under the same condition for the positive percentage of the cells. The percentage difference (|Difference/Average| × 100%) in the positively stained cells between the two outer layers was calculated (MF: Filter side and Air side; PAC: Air side-1 and Air side-2). The average was defined as the average of positively stained cells of the three layers in the tumor slice. Tumor slices with a percentage difference of > 20% were defined as having a gradient for this biomarker. The detailed workflow for determining the biomarker expression gradient was described in Materials and Methods and in Supplementary Figure 2.
The quantifications reported in Figure 2b, 2d, 3b do not support the conclusion that “Loco-regional changes in biomarkers induced in the filter support culture system were overcome with the perfusion air culture (PAC) system”. I understand that the low number and high heterogeneity of samples may result in not-statistically significant differences. Qualitative trends, especially if reproducible, may be acceptable. I recommend to faithfully report all the spatial quantifications for each tumor slices, n=4 for MCF-7, n=3 for H1437, n=15 for OV. Indeed, it is interesting to document the patient heterogeneity of OV samples.
Response: Very good point made by the reviewer. The spatial quantification of biomarker expression was performed for 3 layers of each tumor slice using the method described in last point. The results were shown in the new Figures 2, 3 and 4. Different biomarkers showed different levels of the loco-regional changes. The heterogeneity of ovarian patient samples was higher than the tumors from the CDX. For MCF7 xenograft, the ER and HIF1a expression showed a gradient for all the tumor slices from 6 different xenograft tumors in MF system. For H1437 xenograft, all the analyzed tumor slices from 3 different xenograft tumors in MF system showed a gradient for HIF1a, ɣH2AX and cleaved-Caspase 3 (CC3). Although the patient ovarian tumor slices have a high heterogeneity of the biomarkers expression, we still can observe the highly induced loco-regional changes in MF culture system for some patient tumors. The detailed quantified data of the percentage difference is shown in Supplementary Table 1,2,3.
In the transcriptomic analysis, “Results are from 3 independent experiments for MCF-7 CDX, from 4 independent experiments for H1437 CDX, from 13 independent experiments for primary OV”. What does it mean?
Response: In the transcriptomic analysis, independent experiments were conducted with tumors from 3 different mice of MCF-7 CDX and 4 mice of H1437 CDX. Experiments for primary OV were performed with tumors from 13 different patients.
Are the fold changes of Supplementary Table1 the means of the experiments? If the OV data are very heterogeneous, they should not be pooled together. Please, clarify this point.
Response: In this table gene expression of different treatment conditions were compared. The "logFC" column was calculated after fitting by the limma package. It can be interpreted as an approximation of the difference between means of the log-transformed values in the groups, however, it was not determined directly using the pooled samples.
Page 11. “The primary OV tissues showed a high heterogeneity of cluster patterns already for in vivo samples (Figure 5c), while the MF and PAC systems displayed similar cluster patterns to the in vivo samples”. From what I see, the red points (in vivo) are clearly clustered separately from blue (MF) and green (PAC) points. Therefore, the conclusion does not seem to be correct, but I am not expert in PCA analysis.
Response: Thank you very much for the instruction. Our idea is to illustrate that the patterns of MF and PAC are similar; they cannot be easily separated from each other. In addition, they both remain distinct to red points (in vivo). Meanwhile, the high heterogeneity of these samples may be one possible reason for this phenomena. We have corrected the conclusion in the results part.
About the drug response part, I agree with the conclusion that “The individual drug response can be evaluated in the PAC system”, but the presented data do not support the statements that “the functional response to drug treatment is more sensitive in the PAC system”, nor that “The effects of cisplatin on the tumor slices were higher in the PAC system compared to the MF system”.
Response: We agree with the reviewer´s comment, this statement has been changed in the text.
Figure 9f and 9g only indicate that multiplex staining can be performed on PAC system. The conclusions of lines 516-522 are not supported by the provided images, maybe because of the low quality of PDF file. Quantifications of double/triple positive cells might help. Untreated controls for Figure 9g are missing.
Response: We thank the reviewer for pointing this out. The conclusions in the results part were only preliminary finding we observed during multiplex staining experiments. We have modified the conclusions in the results part. The main purpose of the multiplex staining in this manuscript was to show from technical aspect that:
- This newly developed method can be used for the tumor slices FFPE samples. The thin slices samples were not damaged during the staining process, which involves several rounds of heat-induced epitope retrieval processes.
- After a long-term culture period (up to 8 days) or drug treatment in the PAC system, the co-expression of biomarkers in the tumor slices can still be detected.
The quantification of multiplex staining will be the topic of our next study based on the current manuscript. The untreated controls image was added in the Figure 9g.
Manuscript should be revised to improve clarity. Important information is found in the legends instead than in the main text. For example, the sentence “the HIF1α, γH2AX, Ki67 and CC3 were stained to investigate oxygen supply, DNA damage, proliferation and apoptosis” explains the rational of the experiments and is found only in the legend of Figure 4. The choice of cisplatin is clinically appropriate since this chemotherapy is used for lung and ovarian cancers, but this is never explained. Moreover, only in legend of Figure 6 it is mentioned that H1437 are used as lung cancer model.
Response: We thank the reviewer for this very helpful suggestion. We have added the important information in the results part. The choice of cisplatin used for lung and ovarian cancers was also added in the text and the H1437 as a lung cancer model was mentioned now in the Materials and Methods and results parts.
Did the authors verify the absence of mycoplasma contaminations? Did the authors perform STR validation of cell lines?
Response: All the cell lines were routinely (every 3 month) checked for mycoplasma contamination and the master stock of the cells is authenticated using STR analysis. This information was also added in the Materials and Methods part.
Since this is a methodology paper, it will be important to include more precise information about material and reagents, i.e. both supplier and reference number, for example for filter papers, Matrigel, ATCC and Miltenyi references, media…
Response: We apologize for the lack of some precise information about the material and reagents. All the details were added in the updated version of the manuscript.
Reviewer 2 Report
In „Perfusion air culture of precision-cut tumor slices: an ex vivo system to evaluate individual drug response under precisely controlled culture conditions” Meng Dong and colleagues describe a vertical culture system that uses a holder and cotton meshes for culturing thin tumor tissue slices. They demonstrate that via the capillary structure of cotton meshes the tissue is efficiently supplied from both sides by both, medium and oxygen. With tumor slices from cell-line-derived xenografts (CDX) and primary samples from ovary cancer patients, they further demonstrate that this culture method is equal or in some aspects even better than culturing tissue slices with one side on a membrane and the other at the air – liquid interface. By immunohistochemical staining and high throughput TaqMan-based qPCR tissues were analyzed for the expression of Ki67 and tumor specific markers and compared to non-cultured tissue. Finally, they also analyze the change of the immune population (T cells and macrophages) during culture in this device. Although vertical tissue culture and rinsing tissue slices with a continuous flow of medium as well as the use of cotton gauze is not really new, the combination in this culture holder is a nice and attractive idea for culturing tissue slices. The paper is well written.
Concerns:
In general, I do not believe that PLA and FDM printing are the appropriate materials/methods for printing cell culture devices, as PLA becomes bridle when getting in contact with water and FDM printed structures are hollow, thus capillary forces draw liquids. It is therefore almost impossible to decontaminate them. Especially the procedure of soaking the plastic for 15 min in 70% ethanol seems impropriate, as alcohol will stay in the structure, diffuse out and possibly cause damage to the tissue. Sterilization with a plasma cleaner in a oxygen atmosphere would be a better strategy, if available. So at least it should be stated that these devices are single use (as they are printed at 200°C they might be almost sterile) and for multiple-use devices they should be SLA/DLP printed from resins that are extensively cured and washed before use.
From the description of culture method, it is not clear how the tissues on Millipore filters were cultured, i.e. whether the “air side” was indeed exposed to air. If this is the case, changes in the tissue structure in the region exposed to air are a logical consequence of a rather non-physiologic condition, as (CDX)tissue from solid tumors was used. If this part was not covered with medium, a comparison of the two tissue culture techniques is not really possible. This has to be clarified.
In figure legends, only the number of independent mouse experiments is stated. It is not clear, how many tissue slices per mouse / tumor were analyzed. For example, in Fig 2b the number of data points does not match with the “4 independent experiments” in the figure legend. This has to be addressed.
The same problem with the numbers of tissues slices is also true for Fig 4, where the authors state that 15 primary ovarian cancer cases were analyzed and 9 of them showed a gradient in proliferation but only data of one of them is presented over a time course. What about the other patients and statistics?
In Fig 5 it is not clear for the reviewer, whether in PCA analyses (Fig5c) all different OvCa patients were analyzed (just stated that there were 13 independent experiments? Now 13 patients?). Same also for Fig 6. Please comment!
From Fig 6 it is difficult to say that the presented tissue culture model is really advantageous compared to Millipore filters given the fact that in primary OvCa apoptosis genes were strongly induced.
The experiments on the tumor environment especially on infiltrating T cells are interesting – can you explain, why Treg cells (FoxP3pos) almost disappear from the cultured tissue, whereas other T cell populations and macrophages merely remain constant?
Author Response
Response to Reviewer 2 Comments
In „Perfusion air culture of precision-cut tumor slices: an ex vivo system to evaluate individual drug response under precisely controlled culture conditions” Meng Dong and colleagues describe a vertical culture system that uses a holder and cotton meshes for culturing thin tumor tissue slices. They demonstrate that via the capillary structure of cotton meshes the tissue is efficiently supplied from both sides by both, medium and oxygen. With tumor slices from cell-line-derived xenografts (CDX) and primary samples from ovary cancer patients, they further demonstrate that this culture method is equal or in some aspects even better than culturing tissue slices with one side on a membrane and the other at the air – liquid interface. By immunohistochemical staining and high throughput TaqMan-based qPCR tissues were analyzed for the expression of Ki67 and tumor specific markers and compared to non-cultured tissue. Finally, they also analyze the change of the immune population (T cells and macrophages) during culture in this device. Although vertical tissue culture and rinsing tissue slices with a continuous flow of medium as well as the use of cotton gauze is not really new, the combination in this culture holder is a nice and attractive idea for culturing tissue slices. The paper is well written.
We thank the reviewer for the positive comments on our manuscript.
Concerns:
In general, I do not believe that PLA and FDM printing are the appropriate materials/methods for printing cell culture devices, as PLA becomes bridle when getting in contact with water and FDM printed structures are hollow, thus capillary forces draw liquids. It is therefore almost impossible to decontaminate them. Especially the procedure of soaking the plastic for 15 min in 70% ethanol seems impropriate, as alcohol will stay in the structure, diffuse out and possibly cause damage to the tissue. Sterilization with a plasma cleaner in a oxygen atmosphere would be a better strategy, if available. So at least it should be stated that these devices are single use (as they are printed at 200°C they might be almost sterile) and for multiple-use devices they should be SLA/DLP printed from resins that are extensively cured and washed before use.
Response: We fully agree with the reviewer´s comment. Because of the limitation of our instruments in the lab, we were only able to use the FDM printing when we started this study. We had the same thoughts as the reviewers about the sterilization process. A plasma cleaner in an oxygen atmosphere is definitely a better strategy. Unfortunately, we do not have the instruments for this method. The 3D printing at 210°C should provide a sterile status of the printed chambers, but this process was done outside of the cell culture cabinet and not in a sterilized condition, and we wanted to store the printed chambers for a long period. Therefore, we performed the 15min in 70% ethanol process under the cell culture cabinet, and further stored the printed chambers in the petri dish under sterilized condition. We also considered that the alcohol might stay in the structure, diffuse out and possibly cause damage to the tissue. In order to find a suitable material of 3D printing for the tumor slice culture. We have previously performed experiments to compare different materials (PLA and ABS) for their biocompatibility. We directly used the 70% ethanol treated chambers of both PLA and ABS in the experiments, so we can test the toxicity of the materials and also the toxicity of the possibly remaining ethanol in the materials. We cultured the materials together with a sensitive suspension cell line Mino (mantle cell lymphoma cell line) for 3 days and analyzed the cell viability after cultivation. The results showed that the PLA material and the 15min in 70% ethanol treatment was not toxic to the cells after 3 days of culture (Reviewer Figure 1). The printed PLA chambers were all single used for the experiments. We have added this information about the materials in the results part.
Reviewer Figure 1: Cellular viability of Mino cells after cultivation together with polylactic acid (PLA) or acrylonitrile butadiene styrene (ABS) printed chamber fragments. The chambers were printed by fused deposition modeling (FDM) 3D printing using either PLA or ABS filaments. For sterilization, printed chambers were immersed in 70% ethanol for 15 min, followed by drying out under the cell culture hood. Sterilized chambers were fragmented by hand under the cell culture cabinet. (a) The suspension mantle cell lymphoma cell line Mino was seeded in a 6-well plate at a density of 1.0 x 10^6 cells/ml either alone or together with PLA or ABS chamber fragments. (b) Cellular viability after 3 days as measured by CellTiter-Glo® Luminescent Cell Viability Assay. ATP levels relative to the control condition shown as mean ± SD indicate the viability of the cells (n=3).
From the description of culture method, it is not clear how the tissues on Millipore filters were cultured, i.e. whether the “air side” was indeed exposed to air. If this is the case, changes in the tissue structure in the region exposed to air are a logical consequence of a rather non-physiologic condition, as (CDX)tissue from solid tumors was used. If this part was not covered with medium, a comparison of the two tissue culture techniques is not really possible. This has to be clarified.
Response: We thank the reviewer for this valuable feedback. The details of the Millipore filter culture is shown in Figure 1b. To avoid drying out of the tissue slices, one drop of medium was placed on top of the air-liquid interface (air side) of the tumor slices. The air side was thus covered with a thin layer of medium, so the tumor slices can stay moist without drying out. The detailed description was added in the results parts for Figure 1.
In figure legends, only the number of independent mouse experiments is stated. It is not clear, how many tissue slices per mouse / tumor were analyzed. For example, in Fig 2b the number of data points does not match with the “4 independent experiments” in the figure legend. This has to be addressed.
Response: Thanks for pointing this out. Generally, 15-25 tumor slices can be sliced from each tumor. The obtained tumor slices number depended on the tumor size and condition. Some tumors can also have less than 15 slices or more than 25 slices. We used normally about 4-5 slices for each condition for the experiments. In each condition, out from these 5 slices, 1 slice was used for the IHC staining, 3-4 slices were used for the RNA isolation. Overall, at least 12-15 slices were analyzed per mouse/tumor. This information has been added in the Materials and Methods part.
We have modified the Figure 2 to new figures after new analysis, and the corresponding figure legends were also modified. In old Figure 2b (new Figure 2 c+d), showing the analysis of day 3 samples of MCF7 tumors, 6 independent experiments were analyzed, as mentioned in the middle of the figure legend for Figure (b). In old Figure 2d (new Figure 3b+c), showing the analysis of day 7 samples of MCF7 tumors, 4 independent experiments were analyzed.
The same problem with the numbers of tissues slices is also true for Fig 4, where the authors state that 15 primary ovarian cancer cases were analyzed and 9 of them showed a gradient in proliferation but only data of one of them is presented over a time course. What about the other patients and statistics?
Response: We thank the reviewer for this valuable comment, which is in line with the other referee’s comment. In the old Figure 4 (new Figure 5a), we showed representative images from tumor slices of one patient. In order to show the detailed situation of the other tumors. We have performed the spatial analysis for the IHC images not only for the 15 patient tumor samples but also for the MCF-7 and H1437 CDX tumors. The tumor slices were objectively divided into three layers longitudinally according to the shape of the tumor slices (MF: Air side, Middle, Filter side; PAC: Air side-1, Middle, Air side-2) using the QuPath software (new Figure 2a). All three layers from each tumor slice were analyzed separately under the same condition for the positive percentage of the cells. The percentage difference (|Difference/Average| × 100%) in the positively stained cells between the two outer layers was calculated (MF: Filter side and Air side; PAC: Air side-1 and Air side-2). The average was defined as the average of positively stained cells of the three layers in the tumor slice. Tumor slices with a percentage difference of > 20% were defined as having a gradient for this biomarker. The detailed workflow for determining the biomarker expression gradient is describe in Materials and Methods and in Supplementary Figure 2. The spatial quantification of biomarker expression results were shown in the new Figures 2 and 3 (MCF-7), Figure 4 (H1437) and Figure 5 (patient ovarian tumors). Different biomarkers showed different levels of the loco-regional changes. The heterogeneity of ovarian patient samples was higher than the tumors of the CDX. Although the patient ovarian tumor slices have a high heterogeneity in biomarkers expression, we can still observe the high loco-regional changes induced in the MF culture system for some patient tumors (new Figure 5b). The detailed quantification data of the percentage difference were shown in Supplementary Table 1,2,3.
In Fig 5 it is not clear for the reviewer, whether in PCA analyses (Fig5c) all different OvCa patients were analyzed (just stated that there were 13 independent experiments? Now 13 patients?). Same also for Fig 6. Please comment!
Response: Apologies for the confused expression. In old Figure 5c (new Figure 6c), 13 independent experiments are experiments performed with 13 patient tumors. It is the same for the old Figure 6 (new Figure 7). This was modified in the new figure legends.
From Fig 6 it is difficult to say that the presented tissue culture model is really advantageous compared to Millipore filters given the fact that in primary OvCa apoptosis genes were strongly induced.
Response: Very good point made by the reviewer. From our data in old Figure 6 (new Figure 7), it is shown that different types of tumors responded differently to the culture systems. For the H1437 CDX, the up-regulation of transcripts was predominantly observed for apoptosis in the MF system. Except the apoptosis, other stress-related transcripts were also regulated differently in the MF and PAC systems. On the protein expression level, in old Figure 8b (new Figure 9b), there was no significant difference of the apoptosis marker Cleaved Caspase-3 (CC3) expression between MF and PAC system (MF ctrl vs. PAC ctrl). Due to the high heterogeneity of the primary ovarian tumors, it is hard to draw a conclusion based on the current data from 13 patients. Overall, we can only say that the PAC and MF systems have similar impact on stress gene expression.
The experiments on the tumor environment especially on infiltrating T cells are interesting – can you explain, why Treg cells (FoxP3pos) almost disappear from the cultured tissue, whereas other T cell populations and macrophages merely remain constant?
Response: The very low expression level of FoxP3 in the tumor tissue (maximum 2%) could be one reason. If the biomarker expression level is very low, compared to the big analyzed area of the in vivo tumors, the chance to find the very few positive cells are much smaller in the thin and small tumor slices samples. Other T cell populations and macrophages have relatively high expression level of biomarkers, so the area on the analysis has less influence on the results. To analyze more tumor samples could be one solution for this problem.
Round 2
Reviewer 1 Report
I appreciate the extensive revision work done by the authors. The manuscript is now suitable for publication.
A last very minor point, for the reproducibility of experimental setting, it would greatly help the addition of references or more details about the ‘organotypic supports’: cotton mesh, filter paper and porcine intestine.
Author Response
We thank the reviewer for the positive comments on our revision work of the manuscript and also for the valuable suggestion. We have added the details about the ‘organotypic supports’ in the Materials and Methods (Perfusion air culture system construction part) and in the Results part 3.1.
Reviewer 2 Report
all my questions were addressed - nice study!
Author Response
We thank the reviewer for the positive comments.